# Early Hypocalcemia in Severe Trauma: An Independent Risk Factor for Coagulopathy and Massive Transfusion

**DOI:** 10.3390/jpm13010063

**Published:** 2022-12-28

**Authors:** Marco Vettorello, Michele Altomare, Andrea Spota, Stefano Piero Bernardo Cioffi, Marta Rossmann, Andrea Mingoli, Osvaldo Chiara, Stefania Cimbanassi

**Affiliations:** 1General Surgery and Trauma Team, ASST Niguarda, Milano, Piazza Ospedale Maggiore 3, 20162 Milan, Italy; 2Department of Surgical Sciences, Sapienza University of Rome, Piazzale Aldo Moro 5, 00185 Rome, Italy; 3Department of Medical-Surgical Physiopathology and Transplantation, University of Milan, Festa del Perdono 7, 20122 Milan, Italy

**Keywords:** hypocalcemia, massive transfusion, trauma-induced coagulopathy, trauma

## Abstract

The rapid identification of patients at risk for massive blood transfusion is of paramount importance as uncontrolled exsanguination may lead to death within 2 to 6 h. The aim of this study was to analyze a cohort of severe trauma patients to identify risk factors associated with massive transfusion requirements and hypocalcemia. All major trauma (ISS > 16) presented directly from the scene to the Niguarda hospital between 1 January 2015 and 31 December 2021 were analyzed. A total of 798 patients were eligible out of 1586 screened. Demographic data showed no significant difference between hypocalcemic (HC) and normocalcemic (NC) patients except for the presence of crush trauma, alcohol intake (27% vs. 15%, *p* < 0.01), and injury severity score (odds ratio 1.03, *p* = 0.03). ISS was higher in the HC group and was an independent, even if weak, predictor of hypocalcemia (odds ratio 1.03, *p* = 0.03). Prehospital data showed a lower mean systolic arterial pressure (SAP) and a higher heart rate (HR) in the HC group (105 vs. 127, *p* < 0.01; 100 vs. 92, *p* < 0.001, respectively), resulting in a higher shock index (SI) (1.1 vs. 0.8, *p* < 0.001). Only retrospective studies such as ours are available, and while hypocalcemia seems to be an independent predictor of mortality and massive transfusion, there is not enough evidence to support causation. Therefore, randomized prospective studies are suggested.

## 1. Introduction

Exsanguination and hemorrhagic shock continue to be some of the most preventable causes of death following trauma [1]; therefore, trauma resuscitation is based on early recognition of bleeding, correction, and prevention of factors perpetuating exsanguination such as hypothermia, acidosis, and coagulopathy. While correction of coagulopathy involves delivering high ratios of fresh frozen plasma (FFP) and platelets to packed red blood cells (pBRC), calcium-dependent pathways involved in both intrinsic and extrinsic pathway hemostasis and platelet function are disrupted by bleeding and hemostatic resuscitation itself. While hemorrhage leads to calcium loss, resuscitation with pBRC and FFP contributes to hypocalcemia through the well-known citrate-induced calcium chelation. Other mechanisms justifying hypocalcemia after trauma have been described such as colloid-induced hemodilution [2], acute alcohol intake [3,4], and ischemia-reperfusion following crush injury [5]. Calcium is an important cation in the body and has a fundamental role as a co-factor in enzymatic coagulation reactions such as the formation of fibrin from fibrinogen; the conversion of prothrombin to thrombin; as a cofactor for factors V, VII, VIII, IX, X, and XIII; in the regulation of vasomotor tone and cardiac contractility [6]. A correlation between hypocalcemia and mortality after severe trauma has been demonstrated by several authors and adding hypocalcemia to the infamous triad of death after a traumatic injury has been suggested [7,8]. The primary hypothesis is that there is an association between ionized hypocalcemia, coagulopathy, massive transfusion need, and mortality at hospital discharge. The aim of this cohort retrospective study is to investigate admission ionized calcium levels as a potential predictor of massive transfusion need and the presence of coagulopathy in severe trauma patients. We also tested the hypothesis that hemodilution and acute alcohol intake could be responsible for hypocalcemia in the early phases of severe trauma and tried to assess other potential causative factors of hypocalcemia.

## 2. Materials and Methods

### 2.1. Setting

Niguarda Hospital is the busiest level 1 Trauma Center in northern Italy and admits over 800 adult and pediatric major trauma patients per year (data between 2011 and 2019). It provides major trauma services to an urban area of over 3 million people. Since 2002, a multidisciplinary team (Trauma Team—TT) for the care of the injured has been implemented, and data of all major trauma patients are prospectively collected in a computerized registry. This registry has been updated and periodically checked by TT surgeons. Data are gathered prospectively regarding the amount, type, and time of fluid and blood product transfusion, laboratory test results, vital signs, and other physiologic data. Demographics including age, gender, Glasgow Coma Score (GCS), base excess (BE), and mechanism of injury are also recorded. The institution of the trauma registry for all major trauma admitted to our trauma center has been approved by the Niguarda Ethical Committee Milano Area 3 (record number 534–102,018).

### 2.2. Data

All data were extracted from the hospital trauma registry except for ionized calcium levels that were retrospectively manually reconciled from patients’ records by the authors and incorporated into the final dataset.

### 2.3. Sample

All major trauma that presented directly from the scene to Niguarda hospital between 1 January 2015 and 31 December 2021 (before 2015, ionized calcium was not routinely measured within 30 min from admission to the ER) were eligible for inclusion in the study. The inclusion criteria were an Injury Severity Score (ISS) of ≥16. Patients aged < 18 years, those transferred from another facility, and those who already received intravenous calcium were excluded from the study. We excluded all patients already transfused with pRBC due to citrate-chelation-related hypocalcemia. Regardless of the outcome, patients presenting with out-of-hospital cardiocirculatory arrest were also excluded. Patients missing ionized calcium data from venous blood gas samples at the time of presentation or within 30 min from admission (before any infusion and blood product transfusion) were also excluded. Patients under warfarin or novel anticoagulants were not excluded but their coagulation data were not included in the analysis of the relationship between acute traumatic coagulopathy and hypocalcemia.

### 2.4. Definitions

The threshold for ionized hypocalcemia was defined as <1.11 mmol/L. Ionized calcium was recorded from venous blood gas on patient admission. Severe hypocalcemia was defined as <0.9 mmol/L. Traumatic coagulopathy was defined as INR > 1.5; this threshold was suggested by Peltan [9] as the best predictor for hemorrhagic shock death and overall mortality in trauma patients. Massive transfusion was defined as the transfusion of 10 units of pRBC or more within the first 24 h from admission to the ER. Multiple transfusion was defined as a transfusion between 5 and 10 units of pRBC within the first 24 h. Alcohol intake was considered positive if any alcohol level above 0 mg/dL was found on the blood sample. Prehospital crystalloid infusion volume was analyzed. Colloid infusions were also analyzed although prehospital protocols did not recommend colloid infusions, as there were very few volume expanders and very little data on colloids. The injury severity score (ISS) was calculated as defined by Baker et al. [10]. When the trauma team is activated at Niguarda hospital, a protocol including venous blood gas sampling (BE, ionized calcium, and lactates), and blood sampling for INR, starts at patient admission. Every blood sample in this study was collected prior to any blood product transfusion. Blood gas samples were analyzed with point-of-care testing while INR and aPTT Ratio were processed by the hospital’s laboratory.

### 2.5. Statistical Analysis

Patients were divided into hypocalcemic (HC) and normocalcemic (NC) according to the hypocalcemia definition. Data distribution normality was assessed with the Kolmogoroff–Smirnoff test. Normally or near-normally continuously distributed data were analyzed with the t-Student test and the mean and standard deviation were presented. Non-normal continuous variables were analyzed using the Mann–Whitney U test and the median and 95% confidence intervals were presented. Categorical variables were compared as proportions using the chi-square test or Fisher’s exact test. Associations between variables were studied with univariate regression, and if the *p*-value was <0.05, they were inserted in a multivariable logistic regression to assess their influence on the selected outcome. Independent associations were presented using odds ratios and a 95% confidence interval. Survival rates were displayed using the Kaplan–Meier curve. A *p*-value of <0.05 was considered to be statistically significant. All data were analyzed using StatPlus for Mac version 8 by AnalystSoft Inc., Walnut, CA, USA.

### 2.6. Endpoints

The aims of interest were: to identify hypocalcemia as a potential predictive factor for massive transfusion and coagulopathy; to test the hypothesis that hemodilution and acute alcohol intake can be associated with hypocalcemia on admission and to assess other potential causes of hypocalcemia. In-hospital mortality was also evaluated. 

### 2.7. Statistical Analysis

Patients were divided into hypocalcemic (HC) and normocalcemic (NC) according to the hypocalcemia definition. Data distribution normality was assessed with the Kolmogoroff–Smirnoff test. Normally or near-normally continuously distributed data were analyzed with the t-Student test, and the mean and standard deviation were presented. Non-normal continuous variables were analyzed using the Mann–Whitney U test, and the median and 95% confidence intervals were presented. Categorical variables were compared as proportions using the chi-square test or Fisher’s exact test. Associations between variables were studied with univariate regression, and if the *p*-value was <0.05, they were inserted in a multivariable logistic regression to assess their influence on the selected outcome. Independent associations were presented using odds ratios and a 95% confidence interval. Survival rates were displayed using the Kaplan–Meier curve. A *p*-value of <0.05 was considered to be statistically significant. All data were analyzed using StatPlus for Mac version 8 by AnalystSoft Inc., Walnut, CA, USA.

## 3. Results

A total of 798 patients were eligible out of 1586 screened from our database between January 2015 and December 2021 (Figure 1). A total of 788 patients were excluded from further analysis because of: missing data on ionized calcium (696 patients); transfer from other facilities (29 patients); pre-hospital cardiac arrest (39 patients); pediatric trauma (30 patients); burns (2 patients); do-not-resuscitate decision (1 patient); readmission to ER from ward (1 patient). Hypocalcemia on admission was detected in 16% of patients (Table 1). Demographic data showed no significant difference between HC and NC patients except for the presence of crush trauma, alcohol intake, and injury severity score (Table 2). Patients presenting with crush trauma were all normocalcemic (8 vs. 0, *p* < 0.001).

A higher proportion of patients positive for blood alcohol was found in the HC group (27% vs. 15%, *p* < 0.01). ISS was higher in the HC group and was an independent but weak predictor of hypocalcemia (odds ratio 1.03, *p* = 0.03). Prehospital data showed a lower mean systolic arterial pressure (SAP) and a higher heart rate (HR) in the HC group (105 vs. 127, *p* < 0.01; 100 vs. 92, *p* < 0.001, respectively), resulting in a higher shock index (SI) (1.1 vs. 0.8, *p* < 0.001). No difference was found in the proportion of anticoagulant/antiplatelet therapy between HC and NC patients (*p* = 0.39). A total of 14 patients among the 798 included were on warfarin or new oral anticoagulants. One was hypocalcemic and none needed a massive transfusion (Table 3). They were included in the study population but coagulation data were not included in this analysis. Prehospital crystalloid infusions were significantly different between the two groups, showing a mean of 985 mL in the HC group vs. 710 mL in the NC group, but the difference (200 mL) might not have clinical meaning. Crystalloids were a predictor of HC but with a weak odds ratio of 1.00 (*p* = 0.01). We analyzed the correlation between ionized calcium concentration and the volume of infused crystalloids (Figure 2). The correlation was significant but the slope was not steep (R = −0.254, *p* < 0.01) and $R^{2}$ = 0.06. Hypocalcemic patients have infused colloids in higher volumes than normocalcemic patients (mean 47 vs. 17 mL, *p* < 0.001). Hypocalcemia was associated with coagulopathy (defined by INR > 1.5) (odds ratio 9.47, *p <* 0.001) (Table 4). Hypocalcemia was also associated with massive transfusion need in 25.8% of patients as compared to 3.3% normocalcemic patients (*p* < 0.001), with an odds ratio of 2.42 (*p* = 0.02) and multiple transfusions (51.6% vs. 12.4%, *p* < 0.001, odds ratio 2.18, *p* = 0.007). Mortality was higher in hypocalcemic patients (12.9 vs. 2.7%, *p* < 0.001, odds ratio 2.74 (1.16–6.47), *p* < 0.02 in logistic binary regression) adjusted for age and ISS. Short-term alcohol ingestion was more frequent among hypocalcemic patients (27% vs. 15%, *p* < 0.001) (Table 1) and was independently associated with hypocalcemia when correction for ISS, crystalloids infusion volume, and coagulopathy was applied (Table 4, odds ratio 2.32, *p* < 0.05).

## 4. Discussion

Victims of major traumatic injuries might die from exsanguination within hours of trauma. Many of them present with early coagulopathy at the time of admission to the ER and will require massive transfusions. Early recognition of the need for massive transfusion is of paramount importance for timely resuscitation. Recent studies demonstrated a high incidence of hypocalcemia in traumatic patients ranging from 23% [11] to 56% [1]. While the hypocalcemia definition cutoff may vary between studies, a low ionized calcium level at presentation has been independently associated with coagulopathy defined as INR > 1.5 [9,12]. The importance of calcium in the coagulation cascade, and also in processes involving cardiac muscle contraction in victims of trauma, has been described [6], and its depletion during exsanguination might worsen early traumatic coagulopathy (Table 2). Ionized hypocalcemia was also described in patients suffering from cardiac arrest by Gando et al. [13]. We decided to exclude these patients from our data to avoid the confounding effect of ischemia-reperfusion hypocalcemia due to cardiac arrest. This element, along with our main inclusion criteria (ISS ≥ 16) (including patients without hemorrhagic shock), may explain why hypocalcemia prevalence among our patients (16%) is lower than described in the literature [14]. In our cohort of patients, hypocalcemia was an independent predictor for the need for massive transfusions (MT) (10 red blood packed cells units or more within 24 h) (Table 3) after adjusting for trauma severity (ISS), age, and other confounding effects such as crystalloid infusions discussed later. In our logistic regression model, trauma severity reached statistical significance, while age did not. ISS was a predictor of MT but the hazard ratio was modest (OR 1.07). Lactates were also an independent predictor of MT in our model as also described by Brooke et al. [14] where the authors used a threshold of 4 mmol/L to discriminate between hemodynamically stable trauma patients needing MT or not. Hypocalcemic patients had coagulation parameters showing a consumption coagulopathy (Table 2) and were significantly different from normocalcemic patients, suggesting that hypocalcemia could be tightly related to coagulation. In our hypothesis-generating study, we decided to discuss separately other factors that may cause hypocalcemia, thus worsening hypocalcemia-induced coagulopathy if any.

### 4.1. Crystalloids and Colloids Infusions

We found a difference of 200 mL in the mean amount of crystalloids infused between hypocalcemic and normocalcemic patients (Table 1). Crystalloids were, in fact, a predictor of hypocalcemia in the logistic multivariate analysis (Table 3) but the OR was 1.00 (*p* = 0.01) and the mean difference between the two groups could hardly be held responsible for dilutional hypocalcemia. We analyzed the correlation between ionized calcium concentration and the volume of infused crystalloids (Figure 2). The correlation was significant but the slope was not steep (R = −0.254, *p* < 0.01) and $R^{2}\;$= 0.06. In 2005, Vivien et al. [2] found that dilutional hypocalcemia in trauma patients could be explained by colloids (300 mL ± 300 vs. 1700 mL ± 800 mL) but not by crystalloids infusions (600 mL ± 500 vs. 500 mL ± 500). In our pre-hospital resuscitation protocols, colloids infusions are not recommended, unless no cardiocirculatory stability is reached after crystalloids infusions. This explains why our colloids infusion amounts are lower than what was described by Vivien et al. In our study, although significantly different between HC and NC groups (Table 1), the mean difference in colloid infusions was somehow minimal (47 mL vs. 17 mL) and the colloids infusion amount was not a predictor of hypocalcemia when the logistic multivariate analysis was performed (*p* = 0.41).

### 4.2. Alcohol Intake

Our study showed how acute alcohol intake independently increases the risk for hypocalcemia (odds ratio 2.32, *p* < 0.05). Laitinen et al. [3] explained its effect on hypocalcemia, which showed transient hypoparathyroidism leading to hypocalcemia and hypercalciuria. The additional effect of alcohol intake could then worsen hypocalcemia caused by traumatic injuries.

### 4.3. Crush Injuries

While hypocalcemia should be expected following reperfusion in crush injuries, as a result of calcium uptake by muscle cells [5] and of precipitation of calcium with phosphate released from damaged muscle cells, in this study, patients presenting with crush trauma were all normal or hypercalcemic upon arrival in the ER. Although their number was relatively small to draw conclusions (8 patients, 1% of total traumas studied) and only 2 of them (25%) needed massive transfusions, we should draw particular attention to crush injuries when thinking to use calcium infusions in trauma resuscitation. This retrospective study confirms an independent correlation between early hypocalcemia and the need for massive transfusion in severe trauma patients (OR 2.42). This is not related to the number of infusions administered before ER arrival (colloid infusions were negligible). Hypocalcemia seems, furthermore, worsened by alcohol intake (OR 2.32), but whether this has a role in increasing blood loss after trauma is yet to be determined. Hypocalcemic patients in our cohort were likely to have severe acute coagulopathy (INR > 1.5, OR 9.47) upon arrival adjusted for age, severity score (ISS), and pre-hospital crystalloid and colloid infusions. Hypocalcemia was also an independent predictor for mortality in this cohort (OR 2.74) after adjusting for age and ISS. At hospital discharge, 16 (12.9%) hypocalcemic patients were dead compared to 18 (2.7%) normocalcemic patients (*p* = 0·047). The associations between the length of hospital stay and the timing of death are illustrated in Figure 3.

## 5. Conclusions

In our cohort, hypocalcemia was an independent predictor of massive transfusion need and early coagulopathy as well as mortality following severe trauma, and could play an important role in perpetuating the lethal triad of trauma mortality. As calcium is involved in the coagulation cascade, hemorrhaging patient resuscitation should aim to anticipate or at least correct hypocalcemia. Hypocalcemia after pRBC transfusion is a well-known mechanism and calcium supplementation is included in resuscitation guidelines. Early calcium administration in trauma patients to prevent coagulopathy and reduce transfusion needs, on the contrary, still needs randomized prospective studies to support the causation between early hypocalcemia and coagulopathy itself [15]. Hypercalcemia as defined by ionized calcium > 1.3 resulted in 7 patients (0.9%) in our cohort. In a pilot study from MacKay et al. [16], hypercalcemia (presenting before or as a result of blood transfusions) was associated with higher mortality if compared to normocalcemia. Although a multivariable logistic regression was not performed, due to small patient numbers, the occurrence of hypercalcemia should remain a warning when designing a prospective study on calcium administration for trauma resuscitation.

## Figures and Tables

**Figure 1 jpm-13-00063-f001:**
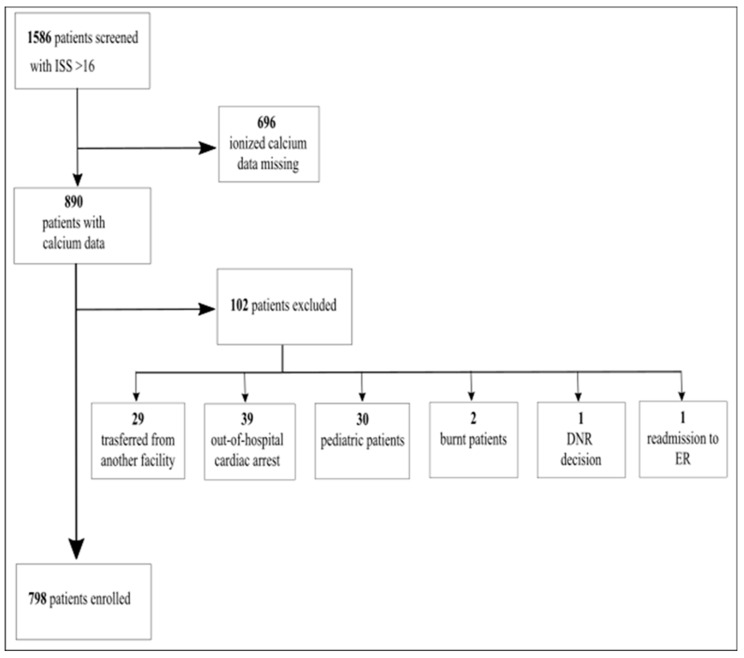
Patient enrollment.

**Figure 2 jpm-13-00063-f002:**
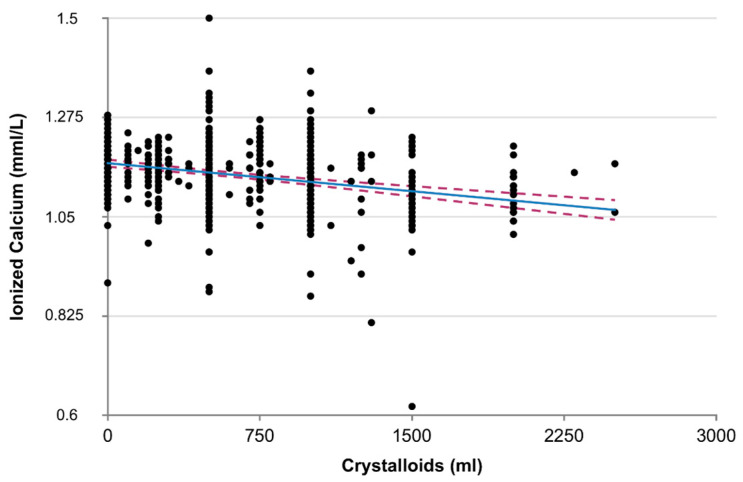
Linear regression: crystalloids infusions vs. ionized calcium. (R = 0.25, R^2^ = 0.065, *p* < 0.001).

**Figure 3 jpm-13-00063-f003:**
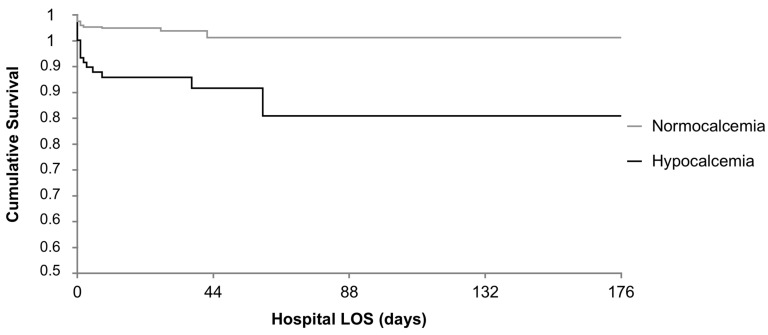
Kaplan–Meier Survival Rates. Hypocalcemic vs. Normocalcemic patients.

**Table 1 jpm-13-00063-t001:** Clinical characteristics of the included patients. (*n*: number of patients, * *p* < 0.01, ** *p* < 0.001, bold: significant).

	Normocalcemia *n* = 669	Hypocalcemia *n* = 129	Total *n* = 798
Characteristics			
Age, yrs, mean +/− SD	47 ± 20	46 ± 20	47 ± 20
Male, *n* (%)	502 (75)	91 (71)	593 (74)
Female, *n (%)*	167 (25)	37 (29)	204 (26)
Blunt trauma, *n (%)*	669 (100)	128 (100)	797 (100)
Penetrating trauma, *n (%)*	0 (0)	0 (0)	0 (0)
Crush trauma, *n (%)*	8 (1)	**0 (0) ***	8 (1)
Isolated Head or Spine trauma, *n (%)*	152 (23)	22 (17)	174 (22)
Alcohol intake, *n (%)*	62 (15) *n* = 409	**22 (27) **** ** *n* ** **= 80**	87 (18) *n* = 489
Anticoagulant/ antiplatelet use, *n(%)*	13 (2) *n* = 625	1 (1) *n* = 120	14 (2) *n* = 745
ISS, median (95% CI)	26 (20–33)	**38 (28–50) ***	26 (21–36)
**Prehospital phase**			
SAP, mmHg, mean ± SD	127 ± 29 *n* = 661	**105 ± 36 *** *n* = 122	123 ± 31 *n* = 783
HR, beats/min, mean ± SD	92 ± 25 *n* = 665	**100 ± 29 *** *n* = 127	93 ± 26 *n* = 792
SI, mean ± SD	0.8 ± 0.3 *n* = 661	**1.1 ± 0.6 *** *n* = 122	0.8 ± 0.4 *n* = 783
Glasgow Coma Scale, median (95%CI)	15 (12–15) *n* = 664	**13 (6–15) ***	15 (11–15) *n* = 792
Cristalloids, mL, mean ± SD	710 ± 456 *n* = 204	**985 ± 509 *** *n* = 56	578 ± 450
Colloids, mL, mean ± SD	7 ± 66 *n* = 662	**47 ± 156 *** *n* = 126	13 ± 88 *n* = 788
**Trauma center admission**			
Hospital SAP, mmHg, mean ± SD	133 ± 28 *n* = 668	**114 ± 29 *** *n* = 124	130 ± 29 *n* = 792
Hospital HR, beats/min, mean ± SD	90 ± 23	**103 ± 26 *** *n* = 125	92 ± 24
Prehospital time, mins, mean ± SD	71 ± 23 *n* = 129	77 ± 24 *n* = 21	72 ± 23 *n* = 150
**Outcome**			
Massive tansfusion, *n* (%)	22 (3.3)	**33 (25.8) ***	55 (6.9)
Multiple transfusions, *n* (%)	83 (12.4)	**66 (51.6) ***	149 (18.7)
pRBC, units, mean ± SD	1.5 ± 3.3	**7.5 ± 9.2***	2.5 ± 5.4
Observed death, *n* (%)	18 (2.7) *n* = 659	**16 (12.9) *** *n* = 124	34 (4.3) *n* = 783

**Table 2 jpm-13-00063-t002:** Biological characteristics of the 798 patients (*n* number of patients, * *p* < 0.01, bold: significant).

	Normocalcemia *n* = 669	Hypocalcemia *n* = 129
Ionized calcium, mmol/L, mean ± SD	1.17 ± 0.05 *n* = 667	**1.03 ± 0.09 ***
Prothrombin time INR, mean ± SD	1.12 ± 0.21 *n* = 576	**1.41 ± 0.4 *** *n* = 103
aPTT ratio, mean ± SD	0.88 ± 0.22 *n* = 573	**1.20 ± 0.63 *** *n* = 102
Base Excess, mmol/L, mean ± SD	−3.5 ± 3.9 *n* = 621	**−8.6 ± 9.8 *** *n* = 125
Lactates, mmol/L, mean ± SD	2.6 ± 1.6 *n* = 663	**4.3 ± 3.3 *** *n* = 127

**Table 3 jpm-13-00063-t003:** Independent association with massive transfusion (logistic multivariate analysis) (bold: significant).

	OR (95% CI)	*p*-Value
ISS	1.07 (1.04–1.10)	**<0.0001**
Lactates, mmol/L	1.29 (1.13–1.48)	**<0.001**
Cristalloids, mL	1.00 (1.00–1.00)	**0.003**
Hypocalcemia (Ionized calcium < 1.1 mmol/L)	2.42 (1.13–5.16)	**0.02**

**Table 4 jpm-13-00063-t004:** Independent association with hypocalcemia (logistic multivariate analysis) (bold: significant).

	OR (95% CI)	*p*-Value
Alcohol intake	2.32 (1.13–4.74)	**0.02**
INR > 1.5	9.47 (2.89–31.0)	**0.0002**
Cristalloids, mL	1.00 (1.00–1.00)	**0.01**
ISS	1.03 (1.00–1.06)	**0.03**

## Data Availability

The data presented in this study are available on request from the corresponding author. The data are not publicly available, to preserve confidentiality.

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
