# Peer review of "Early Hypocalcemia in Severe Trauma: An Independent Risk Factor for Coagulopathy and Massive Transfusion"

_jpm, 2022, doi:10.3390/jpm13010063_

Round 1
Reviewer 1 Report
This is a retrospective review of 798 trauma patients and the relationship between hypocalcemia and mortality as well as massive transfusion. I applaud the authors on tackling a great trauma topic but I do have several questions and comments.
Abstract:
- I'm pretty sure this is the wrong abstract. It has nothing to do with the title of the manuscript nor the rest of the manuscript itself. Please provide the correct abstract.
Introduction:
- Please combine the first couple of sentences and make a full paragraph. These two sentences should not be independent on their own.
- The introduction is a very unorganized collection of thoughts. I encourage the authors to start grand and then narrow down their thinking into the final paragraphs/sentences. I'm not saying their statements are not accurate or important, it just needs a bit more structure as to how you get to the final objective.
- What is your hypothesis for this study? Please provide.
Methods:
- There should not be a period after the 16 in your inclusion criteria sentence.
- How did you determine traumatic coagulopathy was >1.5? Couldn't this just be someone's normal INR? What if they had cirrhosis at baseline?
- How did you address patients on blood thinners?
- Do you use point of care INR or is it processed by the lab?
- I don't understand your outcomes. You state you are looking for risk factors for massive transfusion (in particular the relationship be tween hypocalcemia and massive transfusion), but you are not really looking for risk factors for massive transfuse, only those related to hypocalcemia. Please change the wording and remove the parenthesis for this.
- I am questioning your math for the shock index. Shock index is calculated by heart rate/systolic blood pressure and neither of your numbers make sense. 100/105 and 92/127 are not 1.1 and 0.8 respectively. Please clarify and correct.
- In your tables, you often spell crystalloids with an "i" as in "cristalloids" but then you spell it with a "y" elsewhere. Please correct all to a "y"
Discussion:
- I am still very concerned about acute coagulopathy being set at an INR >1.5. Can you please elaborate on the rationale behind that? I would personally have no problem doing an elective operation on someone with an INR of 1.6 or 1.7 so I have a problem with that being the definition on acute coagulopathy in the trauma patient without additional explanation. Thank you.
- On figure 3, what is all the nonsense on the y-axis? I can't read it and I'm not even sure its relevant. Please correct
Conclusion:
- Although I appreciate your perspective, I do feel like there is an association between hypocalcemia and mortality/massive transfusion. It seems as though you just disregarded all your findings in the conclusion and said more evidence is needed (which I don't disagree and prospective trial is definitely warranted). I would, however, change the wording to better reflect what you found in this study as I do think is it is relevant and needed.
Overall:
- There are several grammatical errors throughout the entire manuscript that need to be corrected.
Author Response
I would like to thank R1 for his suggestions which have deeply improved our paper.
Please find attached the point-to-point letter of response.
I hope that this new R1-enclosed version will be evaluated as suitable for publication on JPM.
Yours sincerely,
Michele Altomare

Reviewer 2 Report
Dear Authors,
hypocalcemia is an underecognized problem in patient with major trauma at risk for massive transfusion.
Your paper highlights the importance of monitoring calcium at admission and during resuscitation of bleeding patients. Moreover your work underlies the importance of predisposing risk factors in the trauma induced coagulopathy. I would suggest checking the content of abstract because it approaches Fourneir's Gangrene instead hypocalcemia in major trauma: probably it was an error.
Author Response
I would like to thank R2 for the suggestions which have deeply improved our paper.
Please find attached the point-to-point letter of response.
I hope that this new R1 enclosed version will be evaluated as suitable for publication on JPM.
Yours sincerely,
Michele Altomare
